# A Scalable Communication Protocol for Networks of Large Language Models

## Abstract

Communication is a prerequisite for collaboration. When scaling networks of AI-powered agents, communication must be versatile, efficient, and portable. These requisites, which we refer to as the *Agent Communication Trilemma*, are hard to achieve in large networks of agents. We introduce Agora, a meta protocol that leverages existing communication standards to make LLM-powered agents solve complex problems efficiently. In Agora, agents typically use standardised routines for frequent communications, natural language for rare communications, and LLM-written routines for everything in between. Agora sidesteps the Agent Communication Trilemma and robustly handles changes in interfaces and members, allowing unprecedented scalability with full decentralisation and minimal involvement of human beings. On large Agora networks, we observe the emergence of self-organising, fully automated protocols that achieve complex goals without human intervention.

## 1 Introduction

Human language evolved primarily for communication purposes (Fedorenko et al., 2024). Despite its inherent ambiguity, natural language provides great versatility and allows humans and machines to collaborate and achieve complex goals that they otherwise could not (Russell & Norvig, 2016).

Decades of literature in computer science explored how to foster collaboration between agents modelled as programs (Wooldridge & Jennings, 1995; Gilbert, 2019). Several research papers design networks of agents to solve complex problems by leveraging each model's specialisation, the so-called rule-based agents paradigm (Wooldridge, 2009). Despite its influence, such a paradigm faces two major limitations: agents hardly adapt to environmental changes and require structured data that limits their versatility (Gilbert & Terna, 2000).

With the advent of Large Language Models (LLM) (Vaswani et al., 2017; Brown et al., 2020), there has been a resurgent interest in networks of collaborative agents. LLMs can solve a variety of problems (Achiam et al., 2023; Dubey et al., 2024a) expressed in natural language as they excel at following instructions (Schulman et al., 2017; Rafailov et al., 2024). LLMs also showed remarkable improvements at handling structured data such as graphs and formatted languages (Kassner et al., 2020; Collins et al., 2022; Jin et al., 2023; Lin et al., 2024).

In terms of performance (e.g., accuracy on classification), the literature suggests that specialised LLMs outperform general purpose models (Hu et al., 2021; Zhang et al., 2024), as well as mitigating the difficulties of handling gargantuan models and the drawbacks of data and model centralisation (Song et al., 2023).

Thus, we hypothesise that:

| Hypothesis |
| --- |
| A network of heterogeneous LLMs can automate various complex tasks with nearly no human supervision via specialised and efficient protocols. |

However, networks of LLM-powered agents face **three key challenges** that make communication at scale significantly more difficult:

- LLMs are **heterogeneous**: different LLMs have different architectures, makers, capabilities and usage policies.[1]
- LLMs are (mostly) **general-purpose** tools: enumerating and standardising each task they can perform is infeasible.
- LLMs are **expensive**: the computational footprint and inference time of "small" LLMs dwarfs that of comparable, specialised APIs.

Scalable communication between heterogeneous LLMs must be **versatile**, i.e., capable of handling a variety of use cases, **efficient**, i.e., requiring the least computational effort, and **portable**, i.e., supporting the protocol should require the least human effort possible. The above-mentioned issues constitute the **Agent Communication Trilemma**, which we expand in Section 3.

In light of this, the aim of this paper is the following:

---

### Key Contribution

We design and implement a communication protocol between heterogeneous LLM-powered agents and assess its feasibility and scalability for solving high-order tasks.

---

We sidestep the Trilemma with **Agora**, a meta protocol that relies on the dual use of structured data for frequent communications and natural language for infrequent ones. With Agora, we instantiate large networks of LLM-powered agents that solve complex tasks autonomously by leveraging efficient communications schemas. In such networks, we observe agents develop **an emergent fully automated protocol to solve a complex task starting from an instruction expressed in natural language**. We believe that this observation can serve as a basis to renew interest in emergent protocols/languages in large networks of LLMs (Lazaridou et al., 2018; Chaabouni et al., 2019; Lazaridou & Baroni, 2020; Chaabouni et al., 2022).

The paper is structured as follows. We first outline the key challenges that constitute the Agent Communication Trilemma (Section 3); we then detail how Agora addresses the Trilemma and serves as a communication protocol for networks of LLMs (Section 4). Finally, in Section 5, we provide two fully functional demos[2]: the former, with two agents, to clarify Agora's operating principles; the latter, with 100, to prove Agora's scalability and show the emergence of self-organising behaviours.

## 2 RELATED WORK

**Multi-agent LLMs and communication.** At the time of writing, Multi-Agent-Systems of Large Language Models (MAS-LLM) have become an active area of research (Guo et al., 2024) after the upsurge of LLMs as general purpose problem solvers (Brown et al., 2020; Achiam et al., 2023; Dubey et al., 2024b). Many fields have adapted techniques from the MAS-LLM paradigm to solve problems single models fail at, including reasoning and math (Li et al., 2024), Theory of Mind (Cross et al., 2024; Li et al., 2023b), planning (Singh et al., 2024), alignment to human values (Pang et al., 2024), and simulation of games, economics, and political scenarios (Bakhtin et al., 2022; Hua et al., 2023; Wu et al., 2024a). The common intuition of these works is that by breaking a task into sub-components (Hong et al., 2023) and allocating a large number of specialised models (Li et al., 2024) to each of them (Li et al., 2023a), one can achieve higher performance and observe emergent behaviours that otherwise would not occur.

On the other hand, a key requisite for solving complex tasks in large networks of MAS-LLMs is effective and efficient communication. In large networks, LLMs must agree on the actions to take (Chen et al., 2023): works such as Agashe et al. (2023) and Liang et al. (2023) studied how LLMs debate to foster collaboration on high-order tasks (Du et al., 2023). Another recent line of

---

[1]Heterogeneity is not unique to agents of LLMs, yet, compared to classic MAS agents, LLMs come with deeper representations of the surrounding environment and are thus more challenging to standardise.

[2]Our code is available anonymously at anonymous.4open.science/r/agora-protocol-demo.

research explores the topology of the MAS-LLM network as a facilitator to reach consensus (Chen et al., 2024).

**LLMs for simulations and emergence of protocols.** A few seminal works studied how emergent communication and protocols arise between neural networks that manipulate symbols (Havrylov & Titov, 2017; Lazaridou et al., 2018; Lazaridou & Baroni, 2020). Written before the rise of LLMs, these works inspired researchers to explore how spontaneous collaboration emerges in MAS-LLMs (Wu et al., 2024b), with application to simulation of *societies* (Gao et al., 2024). Of particular interest for this paper are the works by Chaabouni et al. (2019) and Chaabouni et al. (2022). Chaabouni et al. (2019) describes how emergent communication systems between neural networks privilege longer messages. Chaabouni et al. (2022) posits the existence of "scaling laws" (Kaplan et al., 2020) for large networks of MAS-LLMs in which the dataset, task complexity, and population size are the key to observe emergent behaviours.

## 3 THE AGENT COMMUNICATION TRILEMMA

An agent is a computer system that, in an environment, is capable of autonomous actions (the so-called 'agency' (Horty, 2001)) to meet its design objective (Wooldridge & Jennings, 1995; Wooldridge, 2009, p. 15). Just as humans must negotiate and cooperate to achieve shared goals, so too must agents within multi-agent systems (Wooldridge, 2009, p. 24-25). However, when designing communication protocols for heterogeneous networks (i.e., networks where agents have different architectures, capabilities and design constraints), we run into difficulties when attempting to optimise for three properties at the same time:

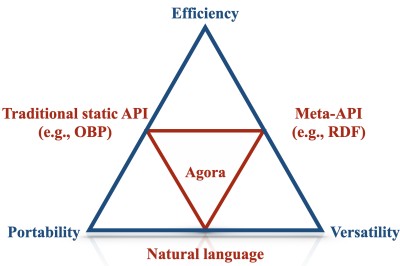

Figure 1: The Trilemma and how our solution (Agora) balances *efficiency*, *portability* and *versatility*.

- **Versatility**: communication between agents should support a wide variety of messages, both in terms of content and format;

- **Efficiency**: the computational cost of running an agent and networking cost of communication should be minimal;

- **Portability**: supporting the communication protocol should require the least implementation effort by the largest number of agents involved.

We name the trade-off between such properties the **Agent Communication Trilemma**, which is illustrated in Figure 1. In the next sections, we will discuss how an LLM-powered communication protocol can trade off versatility, efficiency, and portability.

### 3.1 VERSATILE VS. PORTABLE COMMUNICATION

In networks of agents, versatility and portability are at tension for two fundamental reasons (Olivé, 2007). A prerequisite for two agents who communicate is (1) a shared conceptual understanding of the topic on which they communicate. For instance, two agents can communicate about the weather if they both 'know' what it means to be sunny, rainy and overcast. For example, they should share a similar notion of describing and measuring temperature (e.g., in degrees Celsius). In addition, (2) agents must encode and decode messages in a way that is intelligible for both. Continuing the weather example, if two agents exchange data using JSON objects, both the sender and the receiver must know the syntax (e.g., the keys of a JSON object, such as `temperature`) and the semantics (e.g. `temperature` is a 32-bit floating point value representing the temperature, in central London, as measured in degrees Celsius) of the exchanged messages.

In complex scenarios, defining routines whose syntax and semantics satisfy requisites (1) and (2) may be difficult. For example, a programmer has to manually implement a method to decode (or decode) messages to (or from) other agents. Additionally, the programmer must explicitly instruct the agent about how to manipulate and reason about the message content, often by interpreting API

documentation describing the semantics of the message. Therefore, there is a trade-off between the breadth of messages (**versatility**) and the implementation cost (**portability**).

An example of high-portability, low-versatility is the Open Banking Platform (OBP), which uses a well-defined Open API schema for data transfer (OBL, 2024). OBP is highly portable because it uses a fixed range of well-known concepts which developers can implement; however, it is restricted to discussing a narrow domain of banking data and is thus not versatile. On the other end of the spectrum, rules-based Semantic Web agents (Berners-Lee et al., 2001) that exchange RDF (Beckett et al., 2014) encoded documents are highly versatile since ontologies (Wooldridge, 2009, p. 180) enable the description of structured relations between essentially any concept. Still, they require developers to program agents to implement the specific ontologies used by the network (e.g., if a set of RDF triples states that the temperature is 38°C, an agent must be able to interpret the concepts of "temperature" and "Celsius").

### 3.2 Efficient vs. Versatile and Portable Communication

As previously mentioned, rule-based agents excel at the tasks they are designed to solve but hardly adapt to new environments. Decades of research in reinforcement learning (Sutton, 2018) and then in deep reinforcement learning (Arulkumaran et al., 2017; Henderson et al., 2018), introduced a paradigm where agents learn to optimise their reward as proxy of the task we want them to solve. Agentic-LLMs, i.e., multi-agent systems powered by language models, is a recent paradigm for machine-to-machine communication that relies mostly on their proficiency at handling natural language and following instructions (Li et al., 2023a).

Natural language is highly expressive, making it a suitable choice for versatile communication (Russell & Norvig, 2016). Additionally, LLMs trained on massive corpora seem to develop an implicit understanding of various concepts that **abstracts and makes communication independent from their internal architecture**. Moreover, LLMs can integrate external tools, write code and invoke APIs with relatively little or no training (Schick et al., 2024), since the only requirement is a natural-language description of the tool and its parameters.

Conversely, natural language as a communication medium has two major drawbacks. While engineering and hardware improvements (Dubey et al., 2024b) mitigate costs over time, the computational requirements of invoking an LLM dwarf those of comparable APIs, representing a major bottleneck for scaling networks of LLMs. On the other hand, using closed-source pay-per-usage LLMs hosted by third parties is expensive and raises concerns in terms of replicability of the results (La Malfa et al., 2023). Additionally, natural language is inherently ambiguous: while LLMs have a certain degree of "common sense" to fulfil requests, non-determinism and natural language specifics leave space for errors that routines minimise (for instance, if someone asks for the temperature in Fahrenheit and the agent has a tool that returns the temperature in Celsius, the model must know that Celsius and Fahrenheit are both units of measure for temperature). These factors make LLMs and natural language more prone to errors than other alternatives like handwritten APIs.

In conclusion, RESTful APIs (**efficient**), RDF tuples (**portable**) and natural language (**versatile**) are all trade-offs in the Trilemma. While some approaches are more useful in practice than others, the fact that no communication format achieves all three properties simultaneously suggests that **we need a hybrid communication protocol that leverages all of them**. The next section outlines our solution.

## 4 Agora: a Communication Protocol Layer for LLMs

The key to solving the Communication Trilemma involves accepting that no single protocol can achieve optimal efficiency, portability and versatility at the same time. In this section we introduce Agora, a meta protocol that takes advantage of the unique capabilities of LLMs to *sidestep* the Trilemma by adapting different communications methods for different scenarios.

The most powerful LLMs share three key properties:

- They can understand, manipulate, and reply to other agents using natural language;

(a) An illustration of Agora and how it abstracts the underlying implementation, communication, and physical layers.

(b) Stack of technologies to build Agora.

Figure 2: How Agora fits into a standard communication protocol stack.

- They excel at following instructions, including writing code to implement routines (Schick et al., 2024; Hou et al., 2023; Liu et al., 2024);
- They can autonomously negotiate protocols and reach consensus on strategies and behaviours to adopt in complex scenarios (Chen et al., 2023; Fu et al., 2023).

At its core, Agora uses different communication formats depending on the circumstances; an agent can support a wide breadth of communications (**high versatility**) while handling the majority of the total volume of requests with efficient routines (**high efficiency**). Moreover, the entire negotiation and implementation workflow is handled by the LLMs and requires no human supervision (**high portability**). The concept of protocol documents (PD), which we sketch in Figure 3 and discuss in the next section, lies at the core of Agora's functionalities.

In the next sections, we illustrate the hierarchy of communication methods Agora supports natively and the concept of PD; we then provide an example of how Agora works and how it enables versatile, efficient, and portable communication. We conclude by emphasising how one can integrate and build upon Agora with further technological layers independently from its underlying technologies.

### 4.1 COMMUNICATION IN (AN) AGORA

Agora introduces a machine-readable way to transfer and refer to protocols, namely the protocol documents (PDs). A PD is a plain-text description of a communication protocol.[3] PDs are self-contained, implementation-agnostic, and contain everything an agent needs to support a protocol: this means that most descriptions of existing protocols, such as RFCs, are also suitable PDs. However, instead of relying on a central body to assign identifiers, a PD is uniquely identified by its hash (for multiplexing).

In Agora, the most frequent communications have dedicated efficient routines, and the least frequent ones use inefficient but flexible LLMs and natural language. In particular:

- When possible, frequent communications are handled through traditional protocols, for which there are standard, human-written implementations (e.g., OBP);
- For communications that happen less frequently (or for which there are no standard protocols), agents can use structured data as an exchange medium (which can be handled by LLM-written routines);
- For communications that might be frequent for one side but not the other, the agents still use structured data, but one side can choose to use an LLM, while the other uses a routine;

---

[3]Throughout this paper, we use the word "protocol" to refer to any standardised description of structured communication.

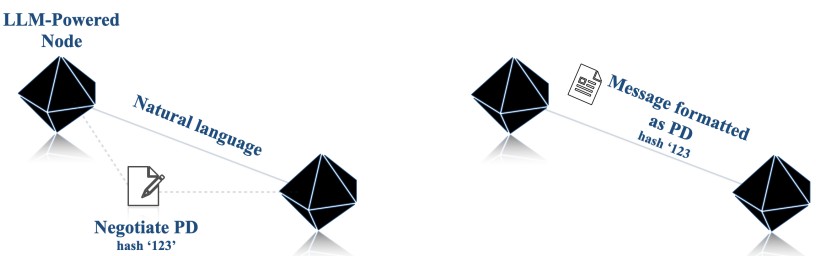

Figure 3: How a protocol document is negotiated between LLM-powered agents (left) and used for future efficient communications.

- For rare communications or when a routine fails unexpectedly, the agents can resort to natural language.

It is entirely up to the agent to handle a query using a human-written routine, an LLM-written routine, or an LLM (or a combination of these three). This gives the agent maximum flexibility over how to process queries.[4] In the Demo (Section 5.3), we will illustrate the trade-off between the versatility of a communication protocol and its expected usage.

Hierarchical communications support any form of communication (**maximum versatility**) , although in practice an LLM is invoked in very rare cases (**maximum efficiency**). Moreover, since LLMs can implement routines on their own (since PDs fully describe the syntax and semantics of a protocol), human programmers only need to provide an overview of the tools the agent has access to, which means that the implementation effort required on the human side is minimal (**maximum portability**). In other words, **Agora sidesteps the Communication Trilemma** by employing routines for frequent requests and resorting to natural language when agents need to negotiate efficient ways to solve a problem or errors occur.

## 4.2 AN EXAMPLE OF COMMUNICATION OVER AGORA

We now describe how two agents, Alice and Bob, can efficiently communicate over Agora using a PD routine, as illustrated in Figure 3. Alice initially sends a query with the hash of its corresponding PD. Bob uses the hash to determine if he has a corresponding routine. If so, he calls it and handles the communication without invoking the LLM. Otherwise, Bob handles the response with the LLM itself.

If Bob uses an LLM to reply several times to queries that follow a given protocol over time, to the point where using an LLM every time becomes expensive, he can use the LLM to write a routine that handles future communications.

If the routine fails or the communication is a one-off instance that does not require a protocol, Alice and Bob use natural language, which is again handled by the LLM. Natural language is also available to bootstrap communication between nodes that have never interacted before, as well as to negotiate new protocols. That said, the lower cost of routines and the lack of ambiguity are strong incentives for agents to prefer structured data.

Note that PDs can be shared with other nodes in the network, which means that two agents that have never interacted before can use protocols developed by other agents.

In the Appendix A, we provide details of five use cases of Agora to further show its versatility as a personal assistant and data analysis tool, and how it leverages compositionality and scalability to reduce costs.

---

[4]Forcing or *nudging* a model to use a specific communication style can improve efficiency, yet its discussion is out of the scope of this paper. One can, for example, specify in the system prompt of an LLM to negotiate a protocol whenever possible.

### 4.3 AGORA AS A LAYER ZERO PROTOCOL

Figure 2 illustrates that Agora is implementation and technology agnostic. The implementation of the agents themselves (e.g., LLMs), the database used to store data (e.g., VectorDB, SQL, MongoDB, etc.), the language in which implementations are written (Python, Java, etc.) and the nature of tools are all abstracted.

At the same time, PDs can refer to other protocol documents, and since routines can call other routines, agents can build upon previous negotiations to solve more complex tasks.

Finally, the versatility and portability of Agora make it straightforward to handle the addition or removal of a node, a change in the capabilities of a node, or a change in the goals of the network, as illustrated in the demo, Section 5.3.

All these factors contribute to making Agora a natural *Layer Zero* protocol, i.e. a foundation layer, for higher-order communication and collaboration between LLMs. We hope our protocol can fuel theoretical and applied research on complex protocols, negotiation schemes, and consensus algorithms in large networks of LLMs.

## 5 AGORA IN PRACTICE

We implement and showcase two scenarios where Agora can be applied. The former, with two agents whose objective is to exchange some data; the latter, with 100, to test Agora scalability and the capacity of LLM-powered agents to autonomously coordinate in complex scenarios. For space reasons, the scenarios are further expanded in Appendices C and D; here, we instead focus on their functionalities and the key observations we drew in terms of **efficiency/versatility/portability**, **reduction of costs**, **scalability** and **emergent behaviours** of fully automated networks of LLMs.

### 5.1 IMPLEMENTATION DETAILS

The design of Agora for our working demos follows three key principles:

- **Minimality.** Agora enforces the basic standards that allow for efficient negotiation and use of protocols, leaving everything else to PDs or other higher-order standards;

- **Decentralisation.** Agora does not rely on central authorities, with any collection of nodes being able to use Agora independently;

- **Full backward compatibility.** Agora supports existing communication protocols and schemas such as OpenAPI and JSON-Schema.

From a practical point of view, Agora uses HTTPS as base communication layer and JSON as format to exchange metadata. When sending a message in a given protocol, an agent sends a JSON document with three keys: the protocol hash, the body of the request formatted according to the protocol, and a non-empty list of sources from which the protocol can be downloaded. The receiver downloads the PD from its preferred source and, upon checking that the hash matches, stores it for future uses. This hash-based identification system ensures that any node can reference any PD without relying on a central authority to assign identifiers. Where PDs are stored is entirely up to the agents; aside from regular cloud storage, hash-based indexing makes decentralised storage options (such as IPFS Benet (2014)) viable. Additionally, since essentially all protocols can be stored as PDs, Agora has full backwards compatibility with existing protocols (although human programmers are encouraged to provide existing, standardised implementations instead of having the LLM re-implement them from scratch).

To simplify negotiation, an agent can expose an endpoint with a list of supported protocols: a potential sender can thus compare the list with its own to automatically determine if there is a common protocol. The sender can also use a potentially unsupported protocol, although the receiver can choose to reject it by returning a predefined error message.

Refer to Appendix B for a more formal description of Agora.

## 5.2 DEMO: RETRIEVING WEATHER DATA

Consider two agents, Alice and Bob. Alice is a Llama-3-405B (Dubey et al., 2024b) powered agent managing the bookings of a guided tour service in London.[5] Bob is a GPT-4o (Achiam et al., 2023) agent for weather service that provides weather forecasts for a given date and location. As part of the user interaction loop, Alice notifies the user if heavy raining is expected on a booked date.

To check the weather, she initially uses her LLM to send a natural language query to Bob (phase `A1`):

> **Alice - Natural Language**
>
> What is the weather forecast for London, UK on 2024-09-27?

Bob uses his Toolformer LLM (Schick et al., 2024) to query his database (phase `B1`) and returns a natural language reply (phase `B2`):

> **Bob - Natural Language**
>
> The weather forecast for London, UK, on 2024-09-27 is as follows:
> "Rainy, 11 degrees Celsius, with a precipitation of 12 mm."

Over time, the cost of invoking an LLM for phases `A1` and `B2` dominate all the other costs; Alice and Bob thus decide to develop a protocol. Alice checks if Bob already supports a suitable protocol but finds none. Therefore, she decides to negotiate a protocol with Bob. After a few rounds of negotiation, Alice and Bob agree on the following protocol: Alice sends a JSON document with two fields, `location` and `date`, and Bob replies with a JSON document containing three fields, namely `temperature` (in degrees Celsius), `precipitation` (in millimetres), and `weatherCondition` (one of "sunny", "cloudy", "rainy" and "snowy"). From there on, Alice specifies the protocol hash when performing a query. An example of exchanged message (excluding Agora's metadata) is:

> **Alice - PD**
>
> `{"location": "London, UK", "date": "2024-09-27"}`

Both Alice and Bob independently decide to write a routine to handle their side of the communication. From now on, Alice and Bob do not need to use the LLM to transmit traffic data: **a routine now automates phases `A1`, `B1` and `B2` and leverages the costs of invoking the respective LLMs**.

**A cost analysis.** In our demo, negotiating the protocol and implementing the routines cost 0.043 USD in API calls, compared to an average cost of 0.020 USD for a natural-language exchange. This means that, as long as Alice and Bob use the agreed-upon protocol more than twice, Agora reduces the overall cost. Please refer to Appendix C for a transcription of the negotiation process and the final protocol.

As a final note, we stress that the entire communication happened without human intervention. Additionally, should Bob become unavailable, Alice can simply reuse the PD with a new node that may use a different LLM/database/technology stack.

## 5.3 DEMO: A NETWORK OF 100 AGENTS

We now show the scaling capabilities and emergent behaviours of Agora by considering a network of 100 LLM-powered agents. In particular, we scale the number of agents, which, as posited

---

[5]While Llama-3 models can be hosted locally, for the sake of a proper comparison with GPT-4o and Gemini, we use a cloud provider, namely SambaNova (https://sambanova.ai).

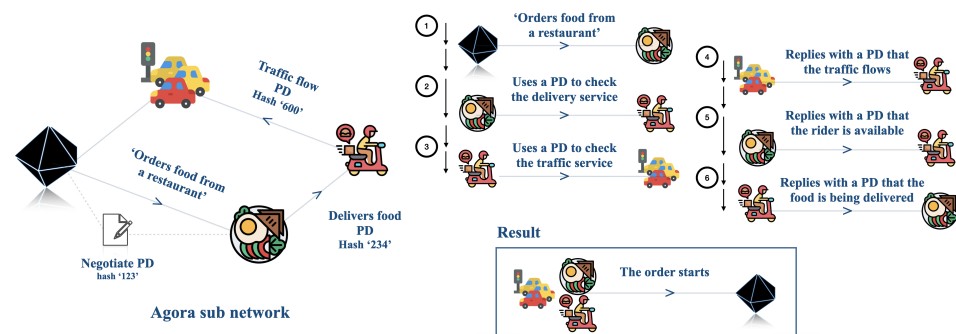

Figure 4: Illustration of how in an Agora network with 100 agents (left; for clarity, only the relevant sub-network is displayed), an emergent protocol for food delivery emerges (right).

in Chaabouni et al. (2022), is a requisite for the emergence of complex behaviours in multi-agent networks.

We design a network of 85 assistant agents interacting with 15 server agents, all powered by LLMs. The server agents offer various services, such as booking hotel rooms, calling taxis, ordering food, etc. An example of a sub-network for food delivery is sketched in Figure 4, left. Their specialisation is handled via prompting, as in Deshpande et al. (2023); Joshi et al. (2023); Li et al. (2023a). As part of their workflow, server agents must interact with several tools and databases; additionally, some servers need to interact with other servers to complete assistants' requests (e.g., taxi services use the traffic data agent to adjust estimated fares for a run). We bootstrap the network by leveraging the underlying communication layer (as described in Section 4 and Figure 2) and inform the nodes of which URLs correspond to which node, as well as manually creating the connection links between agents (e.g. the Taxi Service server knows that the server on port 5007 is a traffic server, but it does not know how to communicate with it and what information it requires);

To showcase the portability of Agora throughout the network, we use different database technologies (SQL and MongoDB) and different LLMs, both open- and closed-source (GPT-4o, Llama-3-405B, and Gemini 1.5 Pro (Reid et al., 2024)). We then generate 1000 random queries, which range from simple ones, such as requesting today's weather, to more complex ones, like booking rooms in ski resorts, buying tickets for movies, ordering one of each dish from a menu, and so on. For each query, assistants receive a JSON document (which represents the task data) and are tasked with fulfilling the request and returning a parsed response that follows a given schema. Queries are distributed among assistants following a Pareto distribution, to simulate some assistants sending significantly more requests than others. Each node can also read and share PDs to one of three protocol databases. Overall, these design decisions result in a very heterogeneous network, testing the limits of Agora. Refer to Appendix D for further implementation details.

**Emergent protocols in large networks.** Once the connections are established and the networks can send and receive messages, we observe several noteworthy behaviours. As PDs are progressively shared between agents (see Figure 5b), we observe the emergence of a decentralised consensus on the appropriate protocols for a given task. An example of this behaviour involves ordering food from restaurants: an agent queries another to request food to be delivered to a certain address. The restaurant agent requests a delivery driver from a food delivery service, who, in turn, checks with the traffic data agent to see if the traffic is smooth enough to fulfil the delivery. None of the agents know each other's roles and the protocols involved beyond their immediate communication. Still, the interaction of the various agents creates an automated workflow that takes care of everything. The emergence of such a protocol is illustrated in Figure 4 (right). In contrast with some recent literature on the emergence of complex protocols (Chaabouni et al., 2019), we observe that with the proper incentives (i.e., efficiency), agents in Agora escape the inefficient *trap* of committing to longer messages in large scale communications.

**A cost analysis.** We compare the cost of running our Agora network against one that uses natural language for all communications. As shown in Figure 5a, at the beginning Agora's cost-efficiency

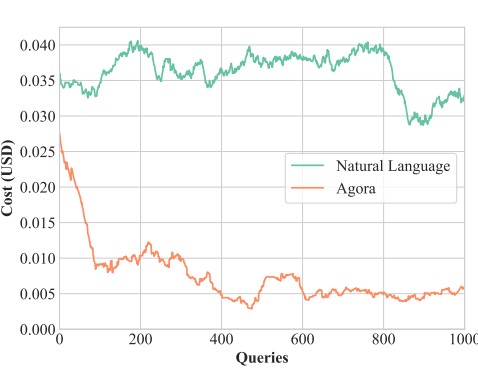

(a) Cost comparison of natural language vs Agora on a network of 100 agents. Costs are averaged with a window size of 100.

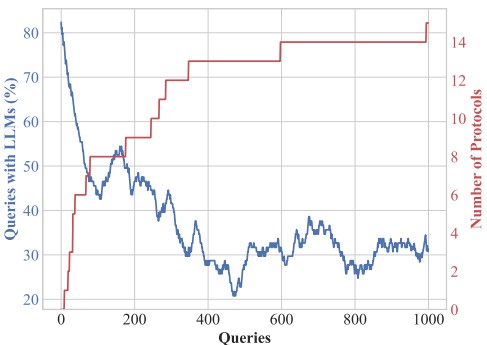

(b) The number of queries to the LLMs in Agora decreases over time as the number of established PDs grows.

Figure 5: Summary of the efficiency of Agora for the demo with 100 agents.

marginally outperforms the network that relies only on natural language; this gap increases over time, with progressively more Agora-powered nodes relying on LLM-written routines. The overall cost in API queries for running 1000 queries in the natural language network is 36.23 USD, compared to Agora's 7.67 USD: in other words, executing this demo with Agora is approximately five times cheaper than with regular natural language. Continuing the demo for more queries would have led to an even larger cost difference.

## 6 CONCLUSIONS

In this paper, we introduced Agora, a meta protocol that sidesteps the Agent Communication Trilemma by using a mix of natural language and structured protocols. We showed that Agora agents can negotiate, implement and use protocols, creating self-organising networks that solve complex tasks. Additionally, we demonstrated the scalability of Agora by testing a 100-agent demo and achieving a five-fold reduction in costs compared to natural language-only communication. Our results showcase the power of negotiation as a basis for efficient, scalable, and decentralised agent networks. As LLMs continue to improve and as interactions between them increase, LLM-powered agent networks have the potential to surpass the scale limitations of single LLMs. Developing frameworks and protocols that enable decentralised, flexible and efficient communication, either through Agora or other technologies, can lay the foundations for a future where complex activities are partially, if not fully, automated by LLMs.

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

# A    AGORA: USE CASES

**S1. Agora as a personal assistant.**

A user is organising a trip to Paris: they want to **book a flight**, **rent a car**, and **book a hotel room**.

The LLM reads the prompt, identifies the actions it has to undertake and checks if there are LLMs available in Agora who can fulfil it. For each service, an LLM is ready to reply.

1. A user sends a message to its personal assistant.

2. The personal assistant dispatches it to Agora.

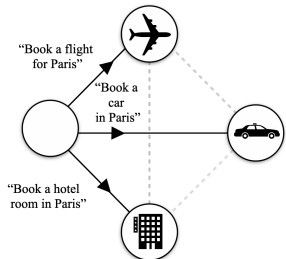

The LLM that acts as personal assistant in the network dispatches the flight, hotel and car requests to the respective LLMs in the network. The messages are dispatched in natural language as there are no pre-existing routines to handle them.

1. The LLM personal assistant dispatches the respective messages to the right node.

2. The car, hotel, and flight LLMs process the requests and turn them into queries for their booking systems.

3. Each LLM replies with their availability and options.

For the next iterations, the LLMs involved in the request propose a routine to standardise the requests to avoid natural language and process the request without invoking the LLMs.

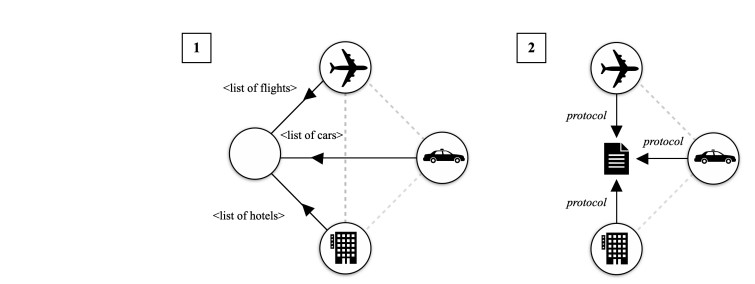

The user receives all the data and decides whether to book or not.

**S2. Security and scalability.**

An LLM (Alice) collects some historical data from another LLM (Bob) that has access to a database **whose internal mechanism and implementation are to keep private**.

Alice submits a request to collect some historical records from Bob. The request is formatted in natural language.

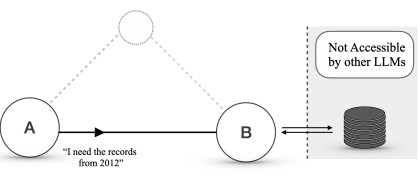

Alice submits another request to Bob.

Bob negotiates a protocol to query its data and writes a shared document protocol in JSON.

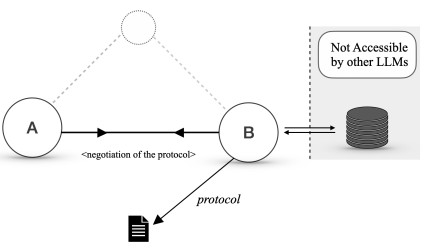

Alice now uses the protocol to query data from Bob.

Bob directly turns the JSON they receive from Alice into a query for its Database.

In this way: Bob **does not invoke the LLM** and the **database internals are not exposed**.

**S3. Compositional tasks.**

An LLM (Alice) wants to (1) analyse some market data and then (2) compute some metrics. Two LLMs in the network can do that.

1. Alice retrieves the protocol documents from a database.

2. Alice finds out that there are two protocol documents that can be used to achieve its goal.

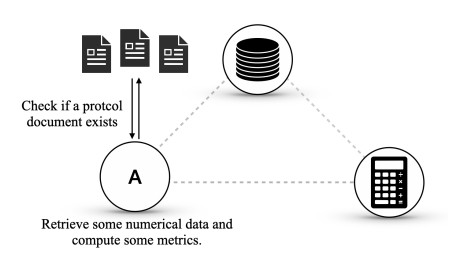

1. Alice submits a request to the first agent to retrieve the data using the first protocol document.

2. Alice receives the data as expected.

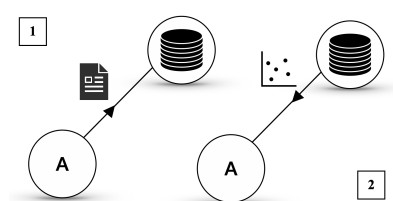

1. Alice submits a request to the second LLM to compute some metrics on the data using the second protocol document.

2. Alice receives the metrics as expected.

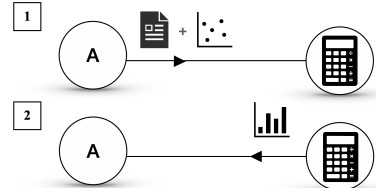

**S4. Scalable consensus in large networks.**

An LLM (Alice) wants to collect and aggregate data points from $N \gg 1$ resources. There is no protocol to handle that, and each resource has its own implementation, possibly not public.

1. Alice submits the requests in natural language.

2. Each queried LLM processes the request, turns it into a routine to retrieve the data and sends it back to Alice.

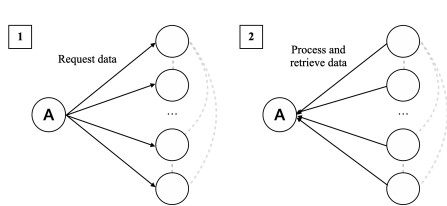

Alice wants to retrieve more data and queries the network another time.

1. One or more receivers suggest using a protocol document for the next iterations.

2. Alice agrees and uses the protocols with as many resources as possible.

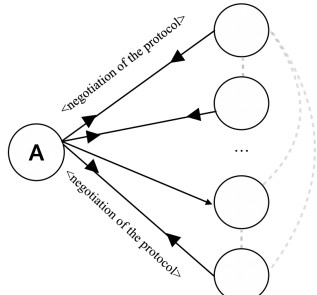

The successive communications will increasingly use protocol documents, thus not necessitating the receiver to process the query with the LLM.

**S5. Scaling complex NLP routines.**

An LLM (Alice) wants to retrieve data from a system powered by an LLM (Bob) that, in turns, obtains its data from a search engine (i.e., the LLM is combined with a RAG). Bob has to (1) turn the natural language request into a query, (2) retrieve the data from the RAG, and (3) return a summary.

Alice queries Bob to retrieve some data. There is no routine to handle any of the three phases, so Bob has to invoke the LLM **twice** to turn the query into a format to invoke the RAG and then perform the summarisation.

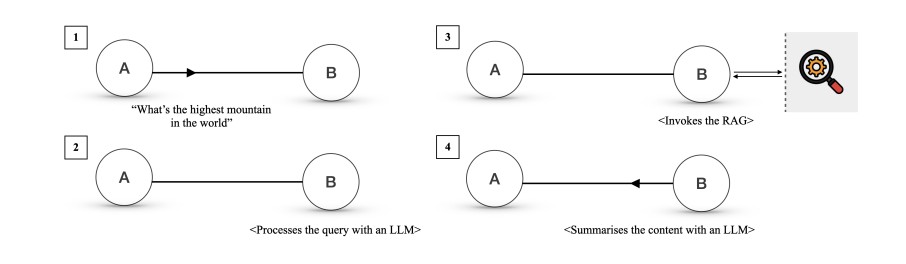

Alice queries Bob again; this time, Bob asks to use a routine to query directly the RAG.

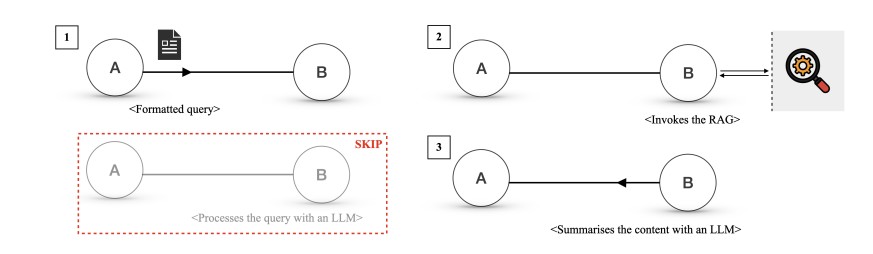

Any query that complies with the document protocol now skips the first phase and directly invokes the RAG.

# B AGORA SPECIFICATION

In this section, we provide a more formal description of Agora.

## B.1 TRANSACTIONS

An Agora transaction operates as follows. Suppose that an agent, Alice, is trying to communicate with another agent Bob:

- Alice sends to Bob over HTTPS a JSON document containing three fields:
  - `protocolHash`: The hash of the protocol document. If natural language is used, then the value of `protocolHash` is `null`;
  - `protocolSources`: A list of URIs where the protocol document can be found. Must be empty if `protocolHash` is `null` and non-empty otherwise;
  - `body`: A string containing the body of the request as specified by the given protocol.
- If Bob does not have the protocol document, he fetches it (either from the sources provided by Alice or from another repository);
- If Bob is unable to use the protocol, he returns a JSON document with one field, namely `status`, which is equal to "rejected";
- Otherwise, Bob computes the response using the LLM, routines, or a combination of both;
- Bob sends as response a JSON document with the following fields:
  - `status`: a string indicating the status of the response (can be "success" or "failure");
  - `body`: the response returned by the agent.
- Note that `"status":"failure"` must be used only for errors that are not covered by the protocol document (e.g., the agent failing to instantiate the LLM); when the protocol prescribes how to handle an error, the agent should return `"status":"success"` and the correct error message as `body`.

## B.2 PROTOCOL DOCUMENTS

A protocol document is identified by its SHA1 hash. Protocol documents can refer to other protocol documents by adding a preamble containing the protocol hash and at least one source for the protocol document. Further discussions concerning the optimal way to reference other protocol documents is left to future work.

## B.3 PREFERRED PROTOCOLS

In order to signal to other agents which protocols are supported by default, an agent can expose an endpoint `/.wellknown`. This endpoint returns a JSON object where each key is a supported protocol hash and each value is a non-empty list of protocol sources.

## C    EXAMPLE OF NEGOTIATION PROCESS BETWEEN TWO AGENTS

We report the negotiation process between Alice and Bob for a protocol to transmit weather forecast. Note that while the LLMs used Markdown-style formatting, for the sake of readability we replaced Markdown syntax with the corresponding typeset style.

---

**Alice**

Hello. I'm here to negotiate a protocol to query a weather forecast service. The service should accept a date and a location as input and return the predicted weather condition, temperature, and precipitation for that day.

Before we start, I'd like to clarify the rules for our negotiation. We should focus on designing a simple and unambiguous protocol that specifies the exact format of the messages sent and received. We shouldn't worry about how the messages will be delivered, and we should assume that the implementation will receive a string and return a string.

To begin, I propose that we use a JSON-based protocol. The input message could be a JSON object with two properties: "date" and "location". The output message could be a JSON object with three properties: "temperature", "precipitation", and "weatherCondition".

What are your thoughts on this proposal?

---

**Bob**

Your proposal aligns well with the goal of keeping the protocol simple and unambiguous. Using a JSON-based protocol for both the input and output messages is a good approach. Here's a more detailed specification for the protocol:

**Input Message** The input message will be a JSON object with the following structure:

```
{
  "date": "YYYY-MM-DD",
  "location": "string"
}
```

- `date`: A string representing the date for which the weather forecast is requested, in the format `YYYY-MM-DD`.
- `location`: A string representing the location for which the weather forecast is requested.

**Output Message** The output message will be a JSON object with the following structure:

```
{
  "temperature": number,
  "precipitation": number,
  "weatherCondition": "string"
}
```

- `temperature`: A number representing the predicted temperature for that day in °C.
- `precipitation`: A number representing the predicted precipitation for that day in mm.
- `weatherCondition` A string representing the predicted weather condition for that day. Possible values are "sunny", "cloudy", "rainy", "snowy".

**Example**
Input:

```
{
  "date": "2023-10-01",
  "location": "New York"
}
```

Output:

```
{
  "temperature": 22.5,
  "precipitation": 5.0,
  "weatherCondition": "cloudy"
}
```

If you agree with this structure, we can finalize the protocol.

**Alice**

LLM response: I agree with the proposed protocol structure. It aligns well with the requirements and is simple and unambiguous.

**Finalised Protocol**

Name: Weather Forecast Query Protocol
Description: A protocol for querying the weather forecast for a given date and location.
**Input Message** The input message will be a JSON object with the following structure:

```
{
  "date": "YYYY-MM-DD",
  "location": "string"
}
```

- `date`: A string representing the date for which the weather forecast is requested, in the format `YYYY-MM-DD`.
- `location`: A string representing the location for which the weather forecast is requested.

**Output Message** The output message will be a JSON object with the following structure:

```
{
  "temperature": number,
  "precipitation": number,
  "weatherCondition": "string"
}
```

- `temperature`: A number representing the predicted temperature for that day in °C.
- `precipitation`: A number representing the predicted precipitation for that day in mm.
- `weatherCondition` A string representing the predicted weather condition for that day. Possible values are "sunny", "cloudy", "rainy", "snowy".

**Example**
Input:

```
{
  "date": "2023-10-01",
  "location": "New York"
}
```

Output:

```
{
  "temperature": 22.5,
  "precipitation": 5.0,
  "weatherCondition": "cloudy"
}
```

# D 100 AGENT DEMO - FULL DESCRIPTION

## D.1 IMPLEMENTATION NOTES

We implemented our demo in Python, using Flask servers as basis for our agents. Each agent is either a *user* or a *server*:

- Users receive a random task, some randomly generated data and a description of the task data (including its schema). Their objective is to execute the requested action and return a reply according to a certain schema. This allows us to generate a large number of queries without needing to handcraft them. Note that all tasks are *single-round*, i.e. they can be fulfilled in one round of communication;
- Servers receive queries from other users and reply to them using a combination of three types of tools:
  - Database tools, which involve connecting to a personal SQL or MongoDB database (assigned at random). Depending on the server, some databases are initialised with dummy data;
  - Mock tools, which are simplifications of actual tools (e.g., for taxi service agents, the `assignTaxi` tool is a mock tool that, instead of actually sending a taxi to a location, mimics the request flow);
  - External tools, which are tools that enable the agent to start a Agora communication with a predefined server, although no information about the respective agents' schema is provided. In other words, the `skiLodge` agent can open a channel with the `weatherService` agent

Moreover, we added three **protocol databases**, which are simple Flask servers that host protocol documents. The first protocol database is a *peer* with the second one, the latter of which is also a peer with the third protocol database (but the first protocol database is not a peer of the third one). Every 10 executed queries, one protocol databases shares its protocol documents with its peers. This simulates the propagation of protocol documents between different databases.

**Picking a Protocol**   Users track the number of communications with a given server about a certain type of task until it hits one of two thresholds: one for using a protocol instead of natural language and one for negotiating a protocol *ex novo*.

When the first threshold is hit, the user invokes the LLM to check if either the server or the reference protocol database (which is randomly assigned to the user at the start of the demo) already have a suitable protocol. If there are none, the user continues using natural language until the second threshold is hit: in that case, the user begins a negotiation with the server and submits the final protocol to the reference protocol database.

Similarly, each server has a counter that tracks the number of natural language communications with *any* user since the last negotiation. Once the counter hits a threshold, the server requests a negotiation with the user, regardless of how many of the tracked queries were sent by the current user. After negotiation, the counter is reset.

In our demo, we set the thresholds for the user to respectively 3 and 5 communications, and the threshold for the server to 10.

**APIs**   For GPT-4o and Gemini 1.5 Pro, we used respectively the OpenAI and Google API. For Llama 3 405b, we used the SambaNova API. Prices per million tokens are reported in Table 1.

**Bootstrapping Quality-of-Life Extensions**   For the sake of bootstrapping the network, while implementing the demo we added two features to our nodes:

- Providing each node with a simple protocol for multi-round communication in natural language;
- Allowing the protocol document to include machine-readable metadata, such as the name or a short description of the protocol. This helps an agent to determine quickly which protocols, among a list of potential protocols, can be suitable for a certain task.

Table 1: Prices per million tokens at the time of writing.

| MODEL | PRICE (USD / 1M TOKENS) | |
| --- | --- | --- |
| | Prompt | Completion |
| GPT-4o | 5.00 | 15.00 |
| Llama 3 405b | 5.00 | 10.00 |
| Gemini 1.5 Pro | 3.50 | 10.50 |

We leave whether these features should be integrated with the Agora standard, or whether they should be handled using PDs only, to future work.

### D.2 EXPERIMENTAL SETUP

**Preliminary Tests** We first ran a series of qualitative tests to determine which among the considered LLMs (OpenAI GPT 4o, Llama 3 405b, Gemini 1.5 Pro) were the most suitable for negotiation and programming. We found that while all three LLMs were capable of negotiating and implementing protocols, GPT 4o was the most robust, followed by the Llama 3 405b and finally Gemini 1.5 Pro. Surprisingly, the main factor behind the brittleness of Gemini 1.5 Pro was not the model's inherent performance, but rather the lack of robustness of the API itself: even with tailored retry systems, the API sometimes failed to respond in a nondeterministic manner (i.e. the same query would at times succeed and at times fail). We believe that our experience was due to temporary server issues, rather than fundamental problems with the model.

**LLM Distribution** In light of our preliminary results, we manually assigned a model to each server node, following a power law consistent with our findings (9 nodes with GPT-4o, 4 nodes with Llama 3 405b, 2 nodes with Gemini 1.5 Pro). User agents were instead randomly assigned one of the three LLMs with uniform distribution. Overall, the breakdown of nodes by model is:

- GPT-4o: 38 nodes (9 server nodes, 29 user nodes)
- Llama 3 405b: 32 nodes (4 server nodes, 28 user nodes)
- Gemini 1.5 Pro: 30 nodes (2 server nodes, 28 user nodes)

Out of 1000 queries, 8 (representing thus 0.8% of the total query volume) failed due to Google's Gemini API not responding. This phenomenon was unrelated to the use of Agora, with 500 Internal Server errors appearing both in the Agora demo and the natural language counterfactual with roughly the same frequency.

**Task Distribution** To simulate the heterogeneity in communication frequency (i.e. how some nodes tend to be more active than others), we assigned to each user a "query budget" (which represents how many queries are sent by a given user) following a Pareto distribution with shape parameter equal to 0.5, adapted so that each user has at least 1 query. The query budget is then split between three randomly chosen types of queries using a Pareto law with a shape parameter of 1 and a minimum of 1 query per type (unless the budget is less than 3 queries). See Figure 6 for a visualisation of the distribution.

### D.3 ADDITIONAL OBSERVATIONS

**Cost Breakdown** The breakdown of cost by activity is as follows:

- Natural language communication: 54%;
- Negotiation: 6%;
- Checking the suitability of existing protocols 22%;
- Implementing the protocols: 17%;

Note that negotiation, despite being the most expensive activity (since it involves several rounds of communication), actually represented the smallest contribution to the total cost, with cheaper but

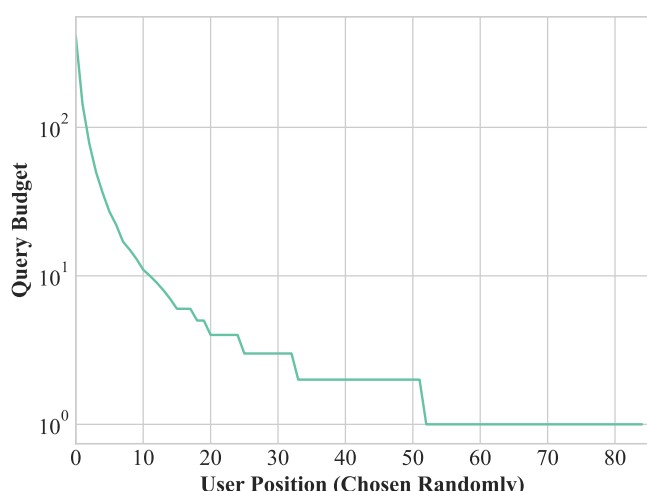

Figure 6: Distribution of query budgets for users. The y axis is logarithmic.

more frequent operations (i.e. sending natural language messages and checking the suitability of protocols) making up the largest portion.

**Similar Protocols**   Due to the (intentional) partial insulation of nodes in the network, sometimes similar protocols emerged independently. Nevertheless, agents using different default protocols were still able to communicate by picking one of the available protocols; for the sake of simplicity, the preferred protocol is chosen by the sender.

