# OpenReview forum: "A Scalable Communication Protocol for Networks of Large Language Models"
_ICLR.cc/2025/Conference — ICLR 2025 Conference Withdrawn Submission_

### Official Review · Reviewer_VEpN · 2024-10-30

**Soundness:** 2
**Presentation:** 2
**Contribution:** 1
**Rating:** 3
**Confidence:** 4

**Summary:**

The authors introduce a communication protocol, called Agora, that use three types of communication for making LLM-powered agents solve complex problems in an efficient way. Specifically, LLM-powered agents use (i) standardized routines for frequent communications, (ii) natural language for rare communications, and (iii) LLM-written routines for communication neither too frequent or too rare. The authors also observe the emergence of self-organizing and fully automated protocols.

**Strengths:**

The domain tackled in the paper is for sure of interest: indeed the emergence of communication modalities and protocols between a group of LLM-based agents is a very valuable topic in the domain of cooperative AI and cooperative LLMs.

**Weaknesses:**

- The main contribution of the paper is the implementation of a communication protocol between heterogenous LLM-based agents. So, for sure an interesting engineering contribution but not a good fit for a machine learning conference;

- The hypothesis is that a network of heterogeneous LLMs can automate a variety of complex tasks. However, this hypothesis does not seem neither so central and neither evaluated seriously and quantitatively in the current version of the paper.

**Questions:**

I think a more detailed explanation/description on how a decentralized consensus on the adequate communication protocol emerge.

Additionally, in which settings and how many times there are inefficiencies or failures in the communication protocols?

---

### Official Review · Reviewer_Shhu · 2024-10-31

**Soundness:** 2
**Presentation:** 3
**Contribution:** 2
**Rating:** 3
**Confidence:** 4

**Summary:**

The paper introduces Agora, a novel communication protocol designed for networks of Large Language Models (LLMs). The authors identify a fundamental "Agent Communication Trilemma" where communication protocols must balance three competing properties: versatility (supporting diverse message types), efficiency (minimal computational cost), and portability (ease of implementation).

This is done by using natural language for infrequent communications while establishing protocols for more frequent communication. These protocols are established through negotiations between agents.

The authors then evaluate Agora in two examples, a network with 2 agents and with 100 agents, and highlight the cost savings of Agora with the 100 agents when compared to only using natural language.

**Strengths:**

Good motivation of the trilemma problem, and timely submission given the interest in agent networks and LLMs. The authors present the trilemma problem in detail (maybe almost too much detail, because other important things then don't have enough space), thus giving a framework to analyse various solutions.

Agora works pragmatically by using efficient protocols for frequent information while using natural language for infrequent requests

Agora does not rely on a single provider for any of the necessary elements

Empirical evaluation of the model in a network with two agents and 100 agents: Showing a significant reduction in the cost of solving the problem

The paper is overall well-written

**Weaknesses:**

No theoretical consideration of the problem: What is the expected cost of various solutions?

Lacking more experimental evaluations:
- Under various network sizes and configurations
- Under various communication distributions.
- Everyone talks frequently to everyone uniformly
- Everyone talks frequently to a few agents
- Only a few agents communicate frequently with a few agents
- Private protocol databases for each agent
- What is the impact of changing the order of the queries?

How large can the networks become? Is it still stable with 1’000 or 10’000 agents?

What is the long-term cost of Agora vs. other methods?
- Generally, missing a comparison to baseline solutions. For instance, a network where the agents can only use protocols, and the other methods the authors mention.
- What is the cost when starting with a partially filled-out database of protocols? Does this save time or only cost?

The paper does not discuss how to handle failures. Natural language can lead to misunderstandings, and the models can make errors when creating the protocols. What happens if there are malicious agents in the system? Can they contaminate the protocol databases?

Missing a measurement of the time complexity of these networks

Unclear of the cost of establishing the protocols is part of the costs

Unclear if the cost only measures the LLM calls. Other APIs

Unclear how/when the agents decide to create a protocol

A lot of the paper is used on the trilemma (and in general text), while the paper is lacking some results in the main paper

I'm generally still quite puzzled how Agora actually works. There must be tons of cases where the agents don't agree on the short hand, or misunderstand each other, etc. All this needs to be discussed in my opinion

I'm willing to raise my grade if I understand better what is actually going on, and what the weaknesses are

**Questions:**

See weaknesses

---

### Official Review · Reviewer_ghNZ · 2024-11-04

**Soundness:** 4
**Presentation:** 4
**Contribution:** 3
**Rating:** 5
**Confidence:** 2

**Summary:**

In this paper, the authors first identify the agent communication trilemma, which refers to the difficulty of achieving versatility, efficiency, and portability in communication. Then the authors propose a communication protocol Agora, which combines standardized routines, natural language, and LLM-written routines to enable efficient problem-solving and self-organization within decentralized networks.

**Strengths:**

1. The identification of the LLM-powered agent communication trilemma.

2. The proposed protocal is novel, which employs a hybrid approach, using standardized routines for frequent communications, natural language for rare communications, and LLM-written routines for everything in between.

3. The paper is well organized and easy to follow.

4. The empirical validation presented is robust, showcasing Agora's scalability and cost-effectiveness through extensive testing involving a network of 100 agents, thereby providing strong evidence for the protocol's practical applicability.

**Weaknesses:**

1. A primary concern regarding the paper is its alignment with the focus of the conference. The content appears to be more closely aligned with themes typically found in communication-focused conferences, which may limit its relevance to the current audience.


2. Some details are missing:
	a. what's the system prompt to guide the negotiation of the protocol in the demo mentioned in section 5.2?

	b. The paper does not provide a thorough analysis of the errors encountered during the demos and how these errors were resolved.


3. In this paper, the models includes are Llama-3-405B which is powerful and has a high-level of instruction-following cability.

	a. How about the models with smaller sizes? How about the agents with varying instruction-following ability?  It is critical to assess the robustness of the Agora protocol across a wider spectrum of model complexities.

	b. Following this inquiry, it remains unclear how the protocol addresses instances where certain LLM models fail to adhere to instructions. Furthermore, the potential for error propagation during communication remains unexamined, which is an important consideration for the protocol’s reliability and efficacy.

**Questions:**

See weakness 1, 2, 3

---

### Official Review · Reviewer_P5zo · 2024-11-05

**Soundness:** 2
**Presentation:** 3
**Contribution:** 1
**Rating:** 3
**Confidence:** 4

**Summary:**

The present work introduces Agora, a meta protocol designed to facilitate communication among heterogeneous LLM agents. It addresses the Agent Communication Trilemma, which posits that scalable communication must be versatile, efficient, and portable. The authors argue that traditional communication paradigms struggle to meet these demands, particularly in networks of LLMs due to their inherent diversity, general-purpose nature, and high computational costs. The proposed Agora aims to integrate standardized routines for frequent communication and natural language for rare interactions.

**Strengths:**

1.Designing efficient communication protocol is an important research problem to scale up multi-agent systems powered by LLMs.

2.The motivation is well-articulated, aligning with current trends in agentic AI and the need for scalable solutions.

3.Research of communication protocol could have broad implications for multi-agent systems.

**Weaknesses:**

1.The empirical experiments are insufficient. Only Figure 5 has some empirical results.

2.The proposed method is not compared against any baseline methods.

3.No theoretical analysis is provided for the communication cost and performance.

Based on the above observations, the present work does not contain enough technical contribution to be published as a research paper. The author may consider to submit as a position paper.

**Questions:**

Please refer to the weaknesses above.

---

### Note · Authors · 2025-01-21

**Comment:**

Dear reviewers,
we have decided to withdraw the paper and restructure it, focusing more on the protocol and infrastructure aspects of our contribution.
We thank you for your feedback.

Best,

The Authors

**Withdrawal Confirmation:**

I have read and agree with the venue's withdrawal policy on behalf of myself and my co-authors.